# Synthesis and Physico-Chemical Properties of Homoleptic Copper(I) Complexes with Asymmetric Ligands as a DSSC Dye

**DOI:** 10.3390/molecules26226835

**Published:** 2021-11-12

**Authors:** Tomohiko Inomata, Mayuka Hatano, Yuya Kawai, Ayaka Matsunaga, Takuma Kitagawa, Yuko Wasada-Tsutsui, Tomohiro Ozawa, Hideki Masuda

**Affiliations:** 1Department of Life Science and Applied Chemistry, Graduate School of Engineering, Nagoya Institute of Technology, Gokiso-cho, Showa-ku, Nagoya 466-8555, Japan; mable.tyo.pai.nmi@gmail.com (M.H.); vip17742@gmail.com (Y.K.); ayaka.matsunaga@rolanddg.co.jp (A.M.); neymneym710@gmail.com (T.K.); wasada.yuko@nitech.ac.jp (Y.W.-T.); ozawa.tomohiro@nitech.ac.jp (T.O.); 2Department of Applied Chemistry, Aichi Institute of Technology, 1247 Yachigusa, Yakusa-cho, Toyota 470-0392, Japan

**Keywords:** dye-sensitized solar cell, homoleptic Cu(I) complex dye, asymmetric ligand

## Abstract

To develop low-cost and efficient dye-sensitized solar cells (DSSCs), we designed and prepared three homoleptic Cu(I) complexes with asymmetric ligands, **M1**, **M2**, and **Y3**, which have the advantages of heteroleptic-type complexes and compensate for their synthetic challenges. The three copper(I) complexes were characterized by elemental analysis, UV-vis absorption spectroscopy, and electrochemical measurements. Their absorption spectra and orbital energies were evaluated and are discussed in the context of TD-DFT calculations. The complexes have high *V*_OC_ values (0.48, 0.60, and 0.66 V for **M1**, **M2**, and **Y3**, respectively) which are similar to previously reported copper(I) dyes with symmetric ligands, although their energy conversion efficiencies are relatively low (0.17, 0.64, and 2.66%, respectively).

## 1. Introduction

Dye-sensitized solar cells (DSSCs) are expected to play an important role as next-generation electricity sources because they have several advantages, such as low production cost and their ability to function under low light intensity [1,2]. However, they also have several disadvantages, such as poor durability and lower conversion efficiency compared to silicon-type solar cells. As a result, research efforts relevant to the improvement of DSSCs have been pursued [3,4,5]. A typical dye used in DSSCs is a ruthenium complex with a polypyridyl ligand, having a high energy conversion efficiency of over 11% [6]. Ruthenium is a relatively expensive transition metal. Therefore, ruthenium-free dyes, such as zinc(II)-based dyes [7,8,9], and organic dyes [10,11,12,13], have been investigated. In recent years, some examples of alternative dyes have been confirmed as having high conversion efficiency comparable to ruthenium-based dyes [9,13]. Copper-based dyes have been investigated because they are much less expensive than ruthenium [14,15,16,17,18,19,20,21,22,23,24,25,26,27,28]. Typical copper(I) complexes tend to be stable under ambient atmosphere due to their tetrahedral structures. The presence of bulky substituent groups tends to enhance the stability of these complexes. The photochemical features of copper(I) complexes are similar to those of ruthenium(II) dyes, which have a strong absorbance contributing to a metal-to-ligand charge transfer (MLCT) [29]. A copper(I) complex dye with mesityl groups has been employed in a DSSC to provide high energy conversion efficiency of 4.66% [24]. Copper complexes have also been used as electrolytes for DSSCs employing their Cu(I/II) redox couples [30,31,32]. Furthermore, DSSCs in which both the dye and the electrolyte are composed of Cu complexes have been reported [33,34]. The two main types of DSSCs based on copper(I) complex dyes are classified as the homoleptic type [14,15,17,18], and the heteroleptic type [16,21,22,24]. These two kinds of dyes are mainly composed of symmetric ligands. The homoleptic type is easily synthesized, but the introduction of distinct functional groups is challenging. The heteroleptic type has two different ligands that possess both functional and anchoring groups on the respective ligands. As a result, a high energy conversion efficiency is expected, but the synthesis of heteroleptic complexes in a solution is also challenging. The use of ligands with large substituents such as mesityl groups is essential in the synthesis of heteroleptic complexes. To solve this problem, a ligand with an attached anchor site is fixed to a surface, and then a Cu complex dye is constructed on the surface [19,23,25,26]. Therefore, the use of asymmetric ligands facilitates the synthesis of homoleptic complexes and creates a coordination environment similar to that of heteroleptic ligands without the use of large substituents. However, the conversion efficiency of DSSCs based on Cu(I) complex dyes with asymmetric ligands reported previously is very low [16]. To address the disadvantages of the previously reported copper(I) dyes, we designed and synthesized new asymmetric ligands, potassium I-2-cyano-3-(6,6′-dimethyl-2,2′-bipyridine-4-yl)acrylate (L1), 2-(3-(4-(dimethylamino)phenyl)-1H-pyrazol-1-yl)-6-methylisonicotinic acid (L2), and 6,6′-dimethyl-2,2′-bipyridine-4-carboxylic acid (L3). The use of asymmetric ligands provides several advantages, such as reducing anchoring groups, routine synthesis, and convenient introduction of the functional groups.

In this study, we designed and synthesized new homoleptic copper(I) complexes with asymmetric ligands containing a cyano group as an electron-withdrawing group, **M1**, a dimethylaminophenyl group as an electron-donating group, **M2**, and lacking a special group, **Y3** (Figure 1). A cyano group is occasionally included in organic dyes [10,11,12,13], to provide DSSCs with high conversion efficiency, accompanied by a red-shifted absorption band [35]. On the other hand, the *N*, *N*-dimethylaniline (DMA) group is used as an electron-donating group [36,37]. The introduction of these functional groups was expected to improve charge injection to oxide semiconductors. We synthesized these three asymmetric ligands, **L1**, **L2**, and **L3** and the copper(I) complexes with an asymmetric ligand, **M1**, **M2**, and **Y3**. Measurements of UV-vis absorption spectra and TD-DFT calculations were carried out to evaluate energy conversion efficiencies. Furthermore, we prepared and evaluated DSSCs based on these complexes.

## 2. Results and Discussion

**Synthesis and characterization of M1, M2, and Y3.** The reactions of copper(II) sulfate and two equivalents of ligands, **L1**, **L2**, or **Y3** accompanied by continuous treatment with ascorbic acid [15] gave homoleptic copper(I) complexes with asymmetric ligands, [Cu(**L1**)_2_]Cl (purple crystal), [Cu(**L2**)_2_]Cl (deep yellow crystal), and [Cu(**L3**)_2_]Cl (dark red crystal), respectively. The preparations of the corresponding copper(I) complexes were characterized by ^1^H NMR, FT-IR, ESI-MS, and elemental analysis.

The ^1^H-NMR spectra of **M1**, **M2**, and **Y3** in CD_3_OD are shown in Appendix A. The ^1^H peaks of the complexes are shifted by the coordination to copper(I) compared to their ligand peaks. Some of the hydrogen peaks of the coordinated ligands are shifted to a lower field region, and the methyl and methylene peaks of the ligands move to an upper field region. The former arises from the coordination of ligands to metal, and the latter results from the ring current effect caused by the approach of the ligand molecules during the formation of homoleptic complexes.

Furthermore, **M1** has a C≡N stretching vibration at 2221 cm^−1^ and a C=O stretching vibration at 1703 cm^−1^, which are not observed in ligand **L1**. For **M2**, the C=N and C=O stretching vibrations are shifted to a low wavenumber region (1611 and 1.716 cm^−1^, respectively) compared with those of ligand **L2**. On the other hand, the C=O stretching vibration of **Y3** is observed at 1718 cm^−1^. Additionally, ESI-MS and elemental analysis clearly indicate the generation of **M1**, **M2**, and **Y3**.

**UV-vis absorption spectra of M1, M2, and Y3.** The UV-vis spectra and data of **M1**, **M2**, and **Y3** are shown in Figure 2a and Table 1, respectively. The UV-vis absorption spectrum of **M1** in EtOH has maximum absorption bands at 497 nm (*ε* = 9330 M^−1^·cm^−1^) and 317 nm (*ε =* 23,200 M^−1^·cm^−1^), which are assigned as the MLCT band (Cu→bpy) and the *π-π*^*^ transition band of bpy, respectively. A similar MLCT band was observed for the previously reported copper(I) bipyridine complex [14,18]. The introduction of an electron-withdrawing cyano group will make a strong absorption band shift to the visible region, as expected for a candidate dye for DSSCs.

For **M2**, the maximum absorption bands are located at 355 nm (*ε* = 25,500 M^−1^·cm^−1^), which overlap with the MLCT band (Cu→py) and *π-π*^*^ transition band (dimethylaminophenyl→py). The MLCT band is significantly blue-shifted relative to that of **M1**. The introduction of electron-donating dimethylamino groups to the ligand results in a blue shift of the absorption band originating from the *π-π*^*^ transition, relative to **M1**. The introduction of pyrazole as a *π*-electron-withdrawing group [38] results in a lowering of the d orbital energy level. As a result, the maximum absorption band originating from MLCT was not identified in the visible region.

On the other hand, the spectrum of **Y3** in EtOH has maximum absorption bands at 475 nm (*ε* = 7480 M^−1^·cm^−1^) and 312 nm (*ε* = 27,500 M^−1^·cm^−1^), which are assigned as the MLCT band (Cu→bpy) and the *π-π*^*^ transition band of bpy, respectively. The spectrum of **Y3** is similar to that of **M1**.

Diffuse reflectance (DR) spectra of **M1**, **M2**, and **Y3** adsorbed on TiO_2_/FTO electrodes, shown in Figure 2b, have broad bands at 491 nm for **M1**, 475 nm for **Y3**, and below 400 nm for **M2**, which are in the same general region as observed in solution (497, 475 and 355 nm for **M1**, **Y3**, and **M2**, respectively) (Figure 2a), indicating that the coordination structures of **M1**, **M2**, and **Y3** are retained on the TiO_2_/FTO electrodes.

**Electrochemical properties of M1**, **M2, and Y3.** To characterize the electrochemical properties of the copper(I) dyes, cyclic voltammetry measurements were carried out in solution and on the TiO_2_/FTO electrode. The reversible and quasi-reversible redox potentials corresponding to Cu(I/II) couple were detected at −0.09 V (*E*_red_/*E*_ox_ = −0.38/0.20 V), 0.33 V (*E*_red_/*E*_ox_ = 0.30/0.37 V), and −0.03 V (*E*_red_/*E*_ox_ = −0.15/0.07 V) vs. Fc/Fc^+^ for **M1**, **M2**, and **Y3** in DMF, respectively, as shown in Appendix A. The potentials are listed in Table 2 together with potentials converted to NHE standard [39,40,41]. In the previous reports, each potential in DMF was converted using *E*_1/2_ (Fc/Fc^+^) = 0.58 V vs. NHE. For **M1**, the CV was measured in the presence of benzimidazole as the protonation agent, because two kinds of redox waves were observed in the absence of benzimidazole. Similar redox potentials were also obtained in the **M1**, **M2**, and **Y3** dyes on the TiO_2_/FTO electrode, as shown in Appendix A, respectively. The values of *E*_1/2_ for **M1**, **M2**, and **Y3** on the TiO_2_/FTO electrode are −0.10 (*E*_red_/*E*_ox_ = −0.39/0.19 V), 0.04 (*E*_red_/*E*_ox_ = −0.35/0.43 V), and −0.05 (*E*_red_/*E*_ox_ = −0.39/0.29 V) vs. Fc/Fc^+^, respectively. The redox potentials of each dye vs. Fc/Fc^+^ and vs. NHE are summarized in Table 2. Each redox potential measured on TiO_2_ in acetonitrile was converted by *E*_1/2_ (Fc/Fc^+^) = 0.62 V vs. NHE [42]. These findings indicate that the properties of the dyes in solution and on the TiO_2_/FTO electrode are similar and are able to endure the structural change induced by the change in charge of the metal ion. However, in the case of **M2**, the reduction wave (Cu^II^→Cu^I^) is significantly shifted to the negative direction, indicating slow electron injection to **M2** on the TiO_2_ surface. The relatively large current observed for **M2** relative to other dyes is likely due to the greater amount of **M2** adsorbed on TiO_2_ (Appendix A). Thus, it appears that **M2** dyes are aggregated on the TiO_2_ surface, resulting in the observation of the different redox behavior on the surface compared with that in solution.

Here, the redox potentials originating from the Cu(I/II) couple generally correspond to the HOMO levels of the dyes. We defined the differences between the *E*_1/2_ values corresponding to Cu(I/II) of the dyes and the redox potential of I^−^/I_3_^−^V vs. NHE as ∆*G*, which indicates a sufficient energy difference for efficient dye regeneration [43,44]. The ∆*G* values estimated for **M1** and **Y3** are low (0.09 and 0.15 V in DMF, respectively), suggesting that the driving force of **M1** and **Y3** dyes that are regenerated by triiodide is small. On the other hand, ∆*G* for **M2** (0.51 V in DMF) is large. This value is similar to that of **N719** in DMF (∆*G* = 0.57 V) [45]. It was previously reported that a ∆*G* value of 0.20–0.25 V is required to provide an efficient dye regeneration process using thiophen-based organic dyes and ferrocene derivatives as redox couples [42]. In the case of the [Co(bpy-pz)_2_]^2+^ redox couple, the ∆*G* value is 0.25 V for triphenylamine-based organic dyes and for the Ru complex dye **D35** [46]. The combination of Ru complex dyes and the I_3_^−^/I^−^ redox couple provides a ∆*G* value ≈ 0.3 V [44]. Furthermore, it has recently been reported that the HOMO level of dyes for DSSCs is required to be near 0.5 V vs. SCE (0.75 V vs. NHE) to provide an efficient electron transfer process [44]. Therefore, the HOMO levels of the dyes will significantly influence the performance of DSSCs. The **M1** and **Y3** complexes used as dyes for DSSCs in this study will cause a slow dye regeneration process. For **M2**, the introduction of pyrazole groups into the ligand might lower the *d* orbital energy level of copper. Indeed, the *E*_1/2_ value corresponding to Cu(I/II) for **M2** was found to be lowered to 0.91 V vs. NHE in the DMF solution. Thus, it was expected that the use of **M2** as a dye for DSSCs might give an optimal regeneration rate.

**DFT calculations.** To understand the absorption spectra and molecular orbital energies of the copper(I) complexes, TD-DFT calculations were performed for **M1**, **M2**, and **Y3**. The calculated absorption spectra and detailed data are shown in Figure 3 and Table 3, respectively. In these calculations, the counter anion Cl^−^, was not considered because it has been reported that the absence of the anion does not generally affect calculation results [17]. For the comparison of the calculated results with the experimental data, the molar extinction coefficients were not added. Instead, we focused on the calculated wavelengths and spectral patterns. In the calculated absorption spectra using B3LYP as the functional, the absorption maxima were found at 665 nm, 469 nm, and 543 nm for **M1**, **M2**, and **Y3**, respectively, which differ significantly from the experimental absorption maxima (497 nm, 355 nm, and 475 nm for **M1**, **M2**, and **Y3**, respectively). However, when CAM-B3LYP was used as the functional, plausible absorption maxima were estimated at 471 nm and 334 nm, and 431 nm for **M1**, **M2**, and **Y3**, respectively. The spectral patterns correspond well with the experimental patterns, as shown in Figure 3 and Table 3, respectively. With this confirmation, the spectral data calculated using CAM- B3LYP were employed for the discussion in this study. The molecular orbital diagram and orbital energy levels evaluated by the calculation using CAM-B3LYP are shown in Appendix A for **M1**, **M2**, and **Y3**, respectively. In the case of **M1**, HOMO−1 and HOMO are localized on the copper atom, and LUMO and LUMO+1 are localized on the pyridine ring containing carboxylate groups. A similar trend was also observed in the case of **Y3**. On the other hand, for **M2**, HOMO–3 and HOMO−2 are localized on the copper atom, HOMO−1 and HOMO are localized to dimethyl aminophenyl groups, and LUMO and LUMO+1 are localized on the pyridine ring containing carboxylate. These observations led us to expect an efficient excited electron transfer from the dyes to TiO_2_. With this expectation, the ratios of the electron distribution of carboxylic acid groups occupied on LUMO and LUMO+1 were estimated. The ratios are 8.8% and 9.3% for **M1**, 20.0% and 20.9% for **M2**, and 12.9% and 12.9% for **Y3**, respectively. This indicates that **M2** is more efficient for the electron transfer to the TiO_2_/FTO electrode than **M1** and **Y3**.

**Photovoltaic performance.** Under AM 1.5 sunlight, the photocurrent density-photovoltage measurements were performed for **M1**, **M2**, and **Y3**. Figure 4 shows I-V curves under light and dark conditions. The details of obtained parameters are given in Table 4. When the DSSC based on **N719** was investigated under the same conditions, we obtained 7.83% as the conversion efficiency. Unfortunately, the energy conversion efficiencies for both dyes are very low. Although **M1** has strong absorption in the visible region, the energy conversion efficiency is very low (0.17 %). This is reflected in the incident photon to current conversion efficiency (IPCE) spectrum (Appendix A). In particular, **M1** has a lower energy conversion (EQE = 0.7% at 480 nm) around the maximum absorption band region (497 nm). This may be due to a smaller driving force for dye regeneration, because it has a small contribution of the molecular orbital at the carboxylic acid site on LUMO and LUMO+1. **M2** also has a low energy conversion efficiency (0.64%). This is rationalized by the fact that it does not have a large absorption band in the visible region. In addition, the amount of **M2** adsorbed on TiO_2_/FTO electrode is quite large (Appendix A) and the adsorption structure of **M2** is probably different. These cause a low conversion efficiency of the DSSC based on **M2**. On the other hand, **Y3** has a large absorption band in the visible region, and its conversion efficiency is relatively high (2.66%). The IPCE data show that the conversion efficiency of the visible area is also relatively high (EQE = 27% at 470 nm) around the maximum absorption wavelength (475 nm). It was also confirmed that the molecular orbitals of the carboxylic acid moiety in **Y3** as the adsorption site on TiO_2_ surface are localized on LUMO and LUMO+1 (Appendix A), resulting in a higher conversion efficiency than other dyes. Unfortunately, the DSSCs based on these dyes might be too inefficient. However, it is interesting that these DSSCs have high *V*_OC_ values (0.48, 0.60, and 0.66 V for **M1**, **M2**, and **Y3**, respectively) in open circuit voltage measurements, which are at the top level among the copper complex-based DSSCs reported hitherto [14,15,16,18,21,22,24]. Provided that for these dyes, the anchoring groups are connected to TiO_2_, the structural change induced by the light excitation will be restrained, which will accelerate the electron transfer to the TiO_2_/FTO electrode and inhibit the reverse electron transfer. FT-IR diffuse reflectance (DR) spectra of copper(I) dyes with asymmetric ligands that were adsorbed on TiO_2_ indicate the absence of free carboxylic acids. However, in the dye portion, it is expected that the two anchoring carboxylic acids are connected to TiO_2_. Therefore, regulation of structural changes may induce promotion of the electron transfer process and inhibition of the backward electron transfer to result in a high *V*_OC_. Among the homoleptic Cu complex dyes with asymmetric ligands, the conversion efficiency of solar cells with **Y3** is relatively high, and it is possible to develop dyes with high *V*_OC_. Since heteroleptic copper complexes are difficult to synthesize, the development of homoleptic copper complexes with high efficiency using asymmetric ligands will be important.

## 3. Materials and Methods

**General.** ^1^H NMR spectra were measured on a Varian Gemini-2000 XL-300 MHz FT spectrometer with TMS as an internal standard. Electronic absorption (UV-vis) spectra were recorded on a JASCO V-550 UV/vis spectrophotometer and diffuse reflection (DR) UV-vis spectra were recorded on a JASCO V-570 spectrophotometer. In the case of DR measurements, the absorption at 800 nm (a region without absorption bands) was subtracted from each spectrum to avoid the effect of differences in reflection coefficients. Infrared absorption spectra were measured on a JASCO FT/IR-410 instrument. Electrospray ionization mass spectra (ESI-MS) were measured using a Micromass LCT ESI-TOF mass spectrometer. Electrochemical measurements in solution were carried out in a three-electrode cell for complexes (0.3 mM) dissolved in dry DMF containing 0.1 M TBAP (tetrabutylammonium perchlorate), a Pt wire used as a counter electrode, and an Ag/Ag^+^ electrode as a reference electrode. After the measurement, ferrocene (Fc) was added as an internal reference. Electrochemical measurements on TiO_2_ were carried out using dyes on TiO_2_/FTO as working electrodes. The dyes on TiO_2_/FTO were prepared by immersing the TiO_2_ electrodes in 0.3 mM EtOH solutions of **M1**, **M2**, **Y3,** and **N719** for about 24 h.

**Chemicals.** All reagents and organic solvents were purchased from Wako Pure Chemical Industries, TCI, Kanto Chemical, MERCK, and Sigma-Aldrich, and were used without further purification. Distilled water was obtained from an EYELA SA-2100E automated water distillation apparatus. **N719** was purchased from Sigma-Aldrich.

**Synthesis.** The ligands **L1**, **L2**, and **L3** were synthesized as carboxylate salts. Homoleptic copper(I) complexes with their ligands, [Cu(**L1**)_2_]Cl (**M1**), [Cu(**L2**)_2_]Cl (**M2**), and [Cu(**L3**)_2_]Cl (**Y3**) were prepared by the reaction of copper(II) sulfate and two equivalents of ligands (**L1**, **L2**, or **L3**), in the presence of ascorbic acid. Detailed synthetic methods for preparing the ligands and their Cu(I) complex dyes are described in the Appendix A.

**Fabrication of Solar cells.** An FTO glass plate was washed in MeCN with an ultrasonic cleaning machine for 30 min. The TiO_2_ paste (PST-18NR) was deposited as an adsorption layer of dye chromophore on the FTO glass plate, which was heated at 100 °C for 10 min. The TiO_2_ paste (PST-400C) was coated as a light scattering layer, and the electrode was sintered at 530°C for 1 h and then immersed in a 0.3 mM EtOH solution of **M1** and **Y3** for 24 h. In the case of **M2**, TiO_2_ electrode was immersed in a 0.3 mM MeOH solution for 24 h. The reference cell was prepared by immersing the plate in a 0.3 mM EtOH solution of **N719** for 24 h. The platinum electrode was prepared by incubating the FTO plate coated with 30 mM 2-propanol solution of H_2_PtCl_6_·6H_2_O at 385 °C for 30 min. The TiO_2_ electrodes covered with dye and Pt electrode were pasted to each other using an epoxy-based adhesive and a UV cured resin. The electrolyte composed of LiI (0.1 M), I_2_ (0.05 M), 1,2-dimethyl-3-propylimidazolium iodide (0.6 M), and 4-tert-butylpyridine (0.5 M) in MeCN was adopted.

**Photovoltaic measurements.** The photovoltaic performance of the DSSCs was assessed using an Asahi Spectra IVP-0605 current-voltage (I–V) curve measurement recorder using an Asahi Spectra HAL-302 solar simulator, which was focused on giving 100 mWcm^−2^, the equivalent of one sun at 1.5 AM at the surface of the test DSSCs. Incident photo-to-current conversion efficiency (IPCE) spectra for DSSCs were recorded with a Newport benchtop optical power meter model 1936-R using an Asahi Spectra PVL-4000 EX3 wavelength-tunable light source. To unify the irradiation area, each DSSC was covered with black tape with a hole (area: 0.07 cm^2^) during measurements to minimize the influence of stray light.

**Evaluation of the amount of each dye on TiO_2_/FTO.** After immersing the TiO_2_/FTO electrode in the EtOH solution of each dye, the adsorbed dye was desorbed as follows. For **N719**: by dipping the electrode into an EtOH/H_2_O (1/1) solution containing 0.1 M NaOH; for **M1** and **Y3**: by dipping the electrode into an EtOH/H_2_O (1/1) solution containing 1 mM NaOH; for **M2**: by dipping the electrode into a MeOH/H_2_O (1/1) solution containing 1 mM NaOH. For Cu complex dyes, 1 mM NaOH aqueous solution was used because of these dyes are unstable under strongly basic conditions. The amount of dye on TiO_2_/FTO was estimated from UV/vis spectra of solutions containing desorbed dye.

**Computational calculations.** The calculations were performed with the Gaussian 09 program package [47]. We carried out all geometry optimizations in a vacuum state by density functional theory (DFT) [48,49,50,51] with B3LYP functional [52]. The following basis sets were used for the respective atoms; 6-311G for Cu, augmented with two p-type polarization functions for the 4p orbitals [53,54], 6-31G(d) [55,56,57] for the other atoms. To assign the absorption bands of electronic transitions in the complexes, time-dependent density functional theory (TD-DFT) [58] calculations were performed using the polarization conductor calculation model (CPCM) [59,60] with B3LYP and the long-range corrected version of B3LYP using the Coulomb-attenuating method (CAM-B3LYP) [61]. In the TD-DFT calculations, the 70 lowest singlet-singlet excitation energies were adopted. We estimated the ratio of carboxylate occupied with any orbitals by using overlap integral [62,63,64,65] and molecular orbitals. The isosurfaces of the molecular orbitals were drawn using MOPLOT and MOVIEW programs [66,67] on the Fujitsu CX400 system at the Nagoya University Information Technology Center. The electronic absorption spectra were calculated using AOMix-package-light [68] on the Windows 7 operating system.

## 4. Conclusions

In our efforts to synthesize Cu(I) complexes that have both advantages of homoleptic- and heteroleptic-type complexes and to overcome their disadvantages using substitutionally active copper(I) ion, we recognized that novel homoleptic complexes with asymmetric ligands should be synthesized. We designed and prepared three homoleptic copper(I) complex dyes with asymmetric ligands, **M1**, **M2**, and **Y3**, which have a small number of anchor groups. Although **M1** has a strong absorption band in the visible region, the energy conversion efficiency of the DSSC based on **M1** was found to be quite low. This might be explained by a small driving force and small contributions of LUMO and LUMO+1 on the carboxylic acid. The energy conversion efficiency of **M2** was also found to be low, which might be due to low light harvesting capability. Otherwise, the introduction of pyrazole groups into the ligand in place of pyridine rings in **M1** (0.62 V vs. NHE), as shown in the CV of **M2** (1.04 V vs. NHE), causes the redox potential of Cu(I/II) to shift to a positive region. This is important from the viewpoint of the dye regeneration process. In the case of **Y3**, its energy conversion efficiency is relatively high (2.66%). We expect that the simple design of the **L3** ligand causes delocalization of LUMO and LUMO+1 at the carboxylic acid as the anchor group, resulting in smooth electron transfer from the dye to the TiO_2_ electrode. It is very interesting that the DSSC based on copper(I) dyes with asymmetric ligands has high *V*_OC_ values (0.48, 0.60, and 0.66 V for **M1**, **M2**, and **Y3**, respectively), although their energy conversion efficiencies are low. This may be caused by the introduction of asymmetric ligands. These findings provide useful insights into the design of new DSSC dyes in the future.

## Figures and Tables

**Figure 1 molecules-26-06835-f001:**
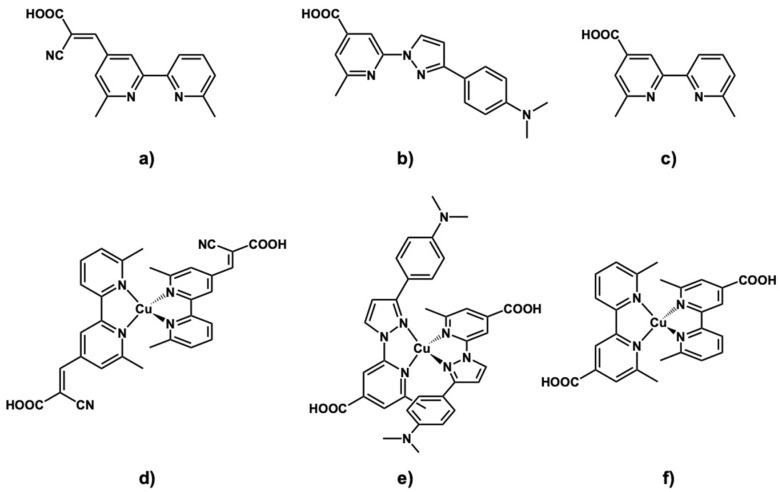
Schematic structures of ligands, (**a**) **L1**, (**b**) **L2**, and **(c) L3**, and their Cu(I) complexes, (**d**) **M1**, (**e**) **M2**, and (**f**) **Y3**.

**Figure 2 molecules-26-06835-f002:**
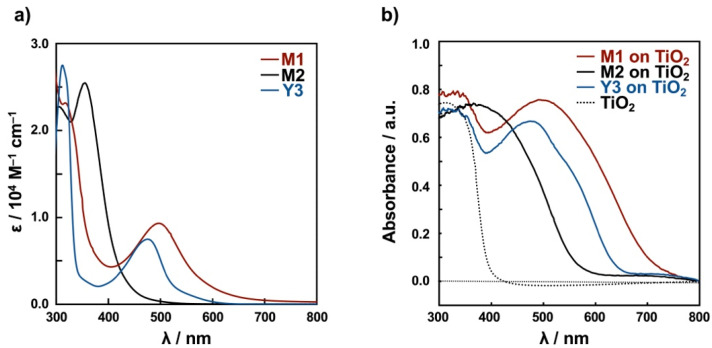
(**a**) UV-vis spectra of **M1**, **M2**, and **Y3** in EtOH and (**b**) their DR spectra on TiO_2_.

**Figure 3 molecules-26-06835-f003:**
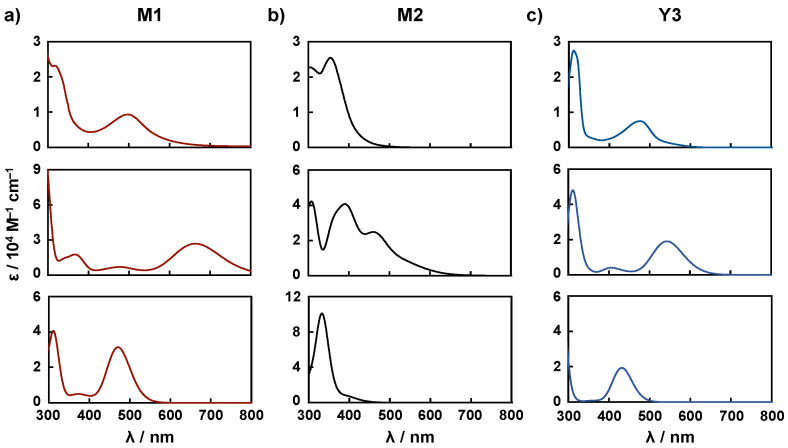
UV-vis absorption spectra of (**a**) **M1**, (**b**) **M2**, and (**c**) **Y3** measured in EtOH (top), as calculated using B3LYP (middle) and CAM-B3LYP (bottom).

**Figure 4 molecules-26-06835-f004:**
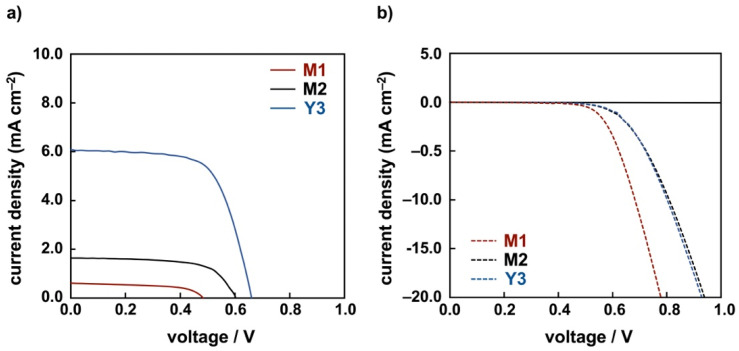
Current-voltage characteristics of dye-sensitized solar cells based on **M1**, **M2**, and **Y3** (**a**) under irradiation and (**b**) in the dark.

**Table 1 molecules-26-06835-t001:** UV-vis and DR spectroscopic data for **M1**, **M2**, and **Y3**.

Dye	in EtOH		on TiO_2_
	*λ*/nm (*ε*/M^−1^·cm^−1^)	Assignment	*λ*_max_/nm
**M1**	317 (23,200) 497 (9330)	π-π* (**L1**) MLCT (Cu(I)→**L1**)	491
**M2**	355 (25,500)	π-π* (**L2**) MLCT (Cu(I)→**L2**)	366
**Y3**	312 (27,500) 475 (7480)	π-π* (**L3**) MLCT (Cu(I)→**L3**)	475

**Table 2 molecules-26-06835-t002:** Redox potentials and Δ*G* values of **M1**, **M2**, and **Y3**.

Dye	in Solution ^1^			on TiO_2_ ^2^		
	*E*_1/2_/V *vs*. Fc/Fc^+^	*E*_1/2_/V *vs*. NHE	Δ*G*/V ^3^	*E*_1/2_/V *vs*. Fc/Fc^+^	*E*_1/2_/V *vs*. NHE	Δ*G*/V ^3^
**M1**	−0.09	0.49	0.09	−0.10	0.52	0.12
**M2**	0.33	0.91	0.51	0.04	0.66	0.26
**Y3**	−0.03	0.55	0.15	−0.05	0.57	0.17

^1^ Scan rate: 0.2 V s^−1^ in DMF. ^2^ Scan rate: 0.01 V s^−1^ in MeCN. ^3^ Δ*G* is defined as the difference in the redox potentials between Cu(I/II) and I^−^/I^3−^.

**Table 3 molecules-26-06835-t003:** UV-vis absorption spectroscopic data for **M1**, **M2**, and **Y3** estimated by DFT calculations.

Functional	Dye	*λ*_calc._ nm	f	Assignment	Character
B3LYP	**M1**	665	0.37	HOMO−1→LUMO (44%) HOMO→LUMO+1 (41%)	MLCT
	**M2**	469	0.23	HOMO−2→LUMO (44%) HOMO–3→LUMO+1 (35%)	MLCT π-π*
	**Y3**	543	0.26	HOMO→LUMO (50%) HOMO−1→LUMO+1 (49%)	MLCT
CAM-B3LYP	**M1**	471	0.43	HOMO→LUMO (23%) HOMO−1→LUMO (22%) HOMO−1→LUMO+1 (21%) HOMO→LUMO+1 (17%)	MLCT
	**M2**	334	0.79	HOMO−1→LUMO+1 (22%) HOMO−2→LUMO (20%) HOMO→LUMO (15%) HOMO−3→LUMO+1 (14%)	MLCT π-π*
	**Y3**	431	0.27	HOMO→LUMO (47%) HOMO−1→LUMO+1 (47%)	MLCT

**Table 4 molecules-26-06835-t004:** Photovoltaic performances of **M1**, **M2,** and **Y3** ^1^.

Dye	*V*_oc_/V	*J*_sc_/mA·cm^−2^	*FF*	*η*/%
**M1**	0.48	0.61	0.58	0.17
**M2**	0.60	1.64	0.65	0.64
**Y3**	0.66	6.08	0.66	2.66
**N719**	0.69	16.5	0.69	7.83

^1^ The electrolyte composed of 0.1 M LiI, 0.05 M I_2_, 0.6 M 1,2-dimethyl-3-propylimidazolium iodide, and 0.5 M 4-*tert*-butylpyridine in MeCN was adopted.

## Data Availability

The data presented in this study is contained within this article and is supported by data in the Appendix A.

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
