# Peer review of "Synthesis and Physico-Chemical Properties of Homoleptic Copper(I) Complexes with Asymmetric Ligands as a DSSC Dye"

_molecules, 2021, doi:10.3390/molecules26226835_

Round 1

Reviewer 1 Report

In this manuscript, Inomata et al. present a series of Cu(I) complexes with bipy-type ligands as well as design of Graetzel cells based thereupon. Overall, the novelty of this work is sufficient for publication, and most of experiments are performed well, so I recommend acceptance after minor revision:

1) The authors mention that they have isolated crystals of all complexes. Are those suitable for SCXRD? It is possible to perform it?

2) The Abstract certainly needs to be re-written - it is rather confusing ("which have both advantages of homoleptic- and heteroleptic-type complexes and compensate their disadvantages...")

3) Please add the ESI-MS spectra to the SI.

Author Response

Thank you very much for your useful comments. The reply for the comments is as below.

Comment: In this manuscript, Inomata et al. present a series of Cu(I) complexes with bipy-type ligands as well as design of Graetzel cells based thereupon. Overall, the novelty of this work is sufficient for publication, and most of experiments are performed well, so I recommend acceptance after minor revision:

Reply: Thank you very much for your positive reviews.

Comment 1): The authors mention that they have isolated crystals of all complexes. Are those suitable for SCXRD? It is possible to perform it?

Reply: The sample was well crystallized, but unfortunately not good enough for single crystal X-ray structure analysis.

Comment 2): The Abstract certainly needs to be re-written - it is rather confusing ("which have both advantages of homoleptic- and heteroleptic-type complexes and compensate their disadvantages...")

Reply: Thank you very much for your useful comments.  According to the comments, we revised the Abstract as follows.  “which have both advantages of homoleptic- and heteroleptic-type complexes and compensate their disadvantages...  “ → “which have advantages of heteroleptic-type complexes and compensate for their synthetic challenges.”

Comment 3): Please add the ESI-MS spectra to the SI.

Reply: Thank you very much for your useful comments.  We added the ESI-MS spectra of three Cu complex dyes as Figure S2.  As the result, the numbering of other figures also revised.

Reviewer 2 Report

This manuscript is not suitable for publication in Molecules for various reasons.

The English is poor with many grammatical errors.

The syntax of many sentences is uncorrected and in some cases the resulting meaning   make no sense: e.g., “The preparations of their copper(I) complexes were well characterized by”.

In other cases, the meaning of some sentences is hardly understandable: e.g., “and are able to endure the structure change by the change in charge of metal ion”; or what do you mean for “the adsorption structure of M2 “?

These are just some of many cases throughout the text. This is clearly unacceptable for a scientific contribution, even if the experimental procedures were correctly done.

Moreover, the resulting performance of these homoleptic Cu(I) complexes as dye-sensitized solar cells is very modest.

Therefore, I cannot support publication of this work.

Author Response

Thank you very much for your helpful comments.  The reply for your comments is as below.

Comment: This manuscript is not suitable for publication in Molecules for various reasons.

The English is poor with many grammatical errors.

The syntax of many sentences is uncorrected and in some cases the resulting meaning make no sense: e.g., “The preparations of their copper(I) complexes were well characterized by”.

In other cases, the meaning of some sentences is hardly understandable: e.g., “and are able to endure the structure change by the change in charge of metal ion”; or what do you mean for “the adsorption structure of M2 “?

These are just some of many cases throughout the text. This is clearly unacceptable for a scientific contribution, even if the experimental procedures were correctly done.

Reply: I am very disappointed with the Reviewer's disappointing response. The English sentences and grammatical errors have been checked and corrected by a native speaker.

Comment: Moreover, the resulting performance of these homoleptic Cu(I) complexes as dye-sensitized solar cells is very modest.

Reply: Thank you for your comments. We believe that your harsh comments are appropriate. The paper we submitted reported on our ligand design strategy. The new ligands were synthesised as designed, although regrettably we did not achieve good performance in terms of conversion efficiency and so on. Unfortunately, for copper complexes, which usually do not show good conversion efficiency, we were able to show reasonable values for Voc, which we consider small but commendable.

Reviewer 3 Report

This is a well done study and decently written manuscript, which could be published in Molecules, when some issues, listed below will be be adressed.

Table 1 the authors probably mean MLCT instead of LMCT?

L150 The authors write: “Thus, the adsorption structure of M2 is probably different from other dyes (…)” What do You mean by adsorption structure?

Could the authors comment on the fact that the concentration of NaOH for M2 desorption is 10 times higher than for other dyes? What was the reason? Considering Figure 2b and similar extinction coefficients 23200-27500 one could expect similar dye load for all dyes instead of about 5 times higher dye load for M2.

In the Figure 2b the spectra are offset, probably due to different reflection coefficients. It is more convenient to compare absorbance values when the offset (the measured value at wavelength where no absorption band is present (e.g.800nm)) is subtracted.

L164 The authors write “It is previously reported that ΔG value of 0.25 V is required for the efficient dye regeneration process.[42] Furthermore, it has recently been reported

that the HOMO level of dyes as DSSC is required to be near 0.5 V vs. SCE (0.75 V vs. NHE)

for an efficient electron transfer process.[44](…)”

Please be more specific, ΔG value of 0.25 applies to I­-/I3- redox mediator.

It is appreciated that the photovoltaic parameters of DSSC comprising studied dyes are referred to N719 efficiencies. I would like to encourage writers to show not only efficiency of reference cells but also Jsc, Voc and FF to give more insightful comparison.

Author Response

Thank you very much for your useful comments.  The reply for your comments is as below.

Comment: This is a well done study and decently written manuscript, which could be published in Molecules, when some issues, listed below will be addressed.

 Reply: Thank you very much for your positive comment, which are a great encouragement to us.

Comment: Table 1 the authors probably mean MLCT instead of LMCT?

Reply: Thank you very much.  This is definitely an MLCT, not an LMCT.  We revised Table 1.

Comment: L150 The authors write: “Thus, the adsorption structure of M2 is probably different from other dyes (…)” What do You mean by adsorption structure?

Reply: Thank you very much for your comments.  Although we have carried out the dye evaluations several times, the amount of M2adsorbed on TiO2/FTO is quite larger compared to those of other dyes.  This suggests that the M2 dye is aggregated on TiO2 surface. We think the aggregation of M2 affects its redox behavior on TiO2/FTO electrode.  We revised the explanation as below:

M2 dyes are aggregated on the TiO2 surface, resulting the observation of the different redox behavior on the surface compared with that in solution.”

Comment: Could the authors comment on the fact that the concentration of NaOH for M2 desorption is 10 times higher than for other dyes? What was the reason? Considering Figure 2b and similar extinction coefficients 23200-27500 one could expect similar dye load for all dyes instead of about 5 times higher dye load for M2.

Reply: Thank you very much for your appropriate comments. We are very sorry that we were wrong about the concentration of NaOH for Cu complex dyes.  All Cu dyes were treated in 1 mM NaOH aqueous solution because of their instability under strong basic conditions.  We revised and added the explanation in “3. Material and Methods” section as below:

“After immersing the TiO2/FTO electrode in the EtOH solution of each dye, the adsorbed dye was desorbed as follows; for N719: by dipping the electrode into an EtOH/H2O (1/1) solution containing 0.1 M NaOH; for M1 and Y3: by dipping the electrode into an EtOH/H2O (1/1) solution containing 1 mM NaOH; for M2: by dipping the electrode into a MeOH/H2O (1/1) solution containing 1 mM NaOH.  For Cu complex dyes, 1 mM NaOH aqueous solution was used because of these dyes are unstable under strongly basic conditions.”

Comment: In the Figure 2b the spectra are offset, probably due to different reflection coefficients. It is more convenient to compare absorbance values when the offset (the measured value at wavelength where no absorption band is present (e.g.800nm)) is subtracted.

Reply: Thank you very much for your comment.  According to reviewer’s comment, Figure 4b was replaced by a revised version and the description of DR measurements has been added to the section “3. Material and Methods” as below:

“In the case of DR measurements, the absorption at 800 nm (a region without absorption bands) was subtracted from each spectrum to avoid the effect of differences in reflection coefficients.

Comment: L164 The authors write “It is previously reported that ΔG value of 0.25 V is required for the efficient dye regeneration process.[42] Furthermore, it has recently been reported that the HOMO level of dyes as DSSC is required to be near 0.5 V vs. SCE (0.75 V vs. NHE) for an efficient electron transfer process.[44](…)”  Please be more specific, ΔG value of 0.25 applies to I­-/I3- redox mediator.

Reply: Thank you very much for your useful comments.  According to your comments, we added more detailed explanation about ∆Gand added the new reference (ref. 46) as below and the numbering of the references after ref. 47 was revised:

“It was previously reported that a ∆G value of 0.20 ~ 0.25 V is required to provide an efficient dye regeneration process using thiophen-based organic dyes and ferrocene derivatives as redox couples.[42]  In the case of [Co(bpy-pz)2]2+ redox couple, the ∆G value is 0.25 V for triphenylamine-based organic dyes and for the Ru complex dye D35.[46]  The combination of Ru complex dyes and I3/I redox couple provides a ∆G value ≈ 0.3 V.[44]”

Comment: It is appreciated that the photovoltaic parameters of DSSC comprising studied dyes are referred to N719 efficiencies. I would like to encourage writers to show not only efficiency of reference cells but also Jsc, Voc and FF to give more insightful comparison.

Reply: Thank you very much for your helpful comments.  According to your comments, we added the photovoltaic parameters for N719 in Table 4 and revised its footnote.

Round 2

Reviewer 2 Report

Even if the authors didn’t reply to all my comments, the revised manuscript is certainly improved and, hence, suitable for publication in Molecules.